# Quantification of hs-Troponin Levels and Global Longitudinal Strain among Critical COVID-19 Patients with Myocardial Involvement

**DOI:** 10.3390/jcm13020352

**Published:** 2024-01-08

**Authors:** Mochamad Yusuf Alsagaff, Louisa Fadjri Kusuma Wardhani, Ricardo Adrian Nugraha, Tony Santoso Putra, Bagus Putra Dharma Khrisna, Makhyan Jibril Al-Farabi, Ruth Irena Gunadi, Yusuf Azmi, Christian Pramudita Budianto, Rosi Amrilla Fagi, Nadya Luthfah, Agus Subagjo, Yudi Her Oktaviono, Achmad Lefi, Budi Baktijasa Dharmadjati, Firas Farisi Alkaff, Budi Susetyo Pikir

**Affiliations:** 1Department of Cardiology and Vascular Medicine, Faculty of Medicine Universitas Airlangga—Dr. Soetomo General Hospital, Jalan Mayjend Prof. Dr. Moestopo 6-8, Surabaya 60286, Indonesia; louisa.fadjri.kusumawardhani-2017@fk.unair.ac.id (L.F.K.W.); ricardo.adrian.nugraha-2019@fk.unair.ac.id (R.A.N.); tony.santoso.putra-2019@fk.unair.ac.id (T.S.P.); bagus.putra.dharma-2019@fk.unair.ac.id (B.P.D.K.); makhyan.jibril.al-2018@fk.unair.ac.id (M.J.A.-F.); ruth.ena.gunadi-2018@fk.unair.ac.id (R.I.G.); yusuf.azmi-13@fk.unair.ac.id (Y.A.); christian.pramudita@fk.unair.ac.id (C.P.B.); rosi.amrilla@fk.unair.ac.id (R.A.F.); nadya.luthfah@fk.unair.ac.id (N.L.); agus.subagjo@fk.unair.ac.id (A.S.); yudi.her@fk.unair.ac.id (Y.H.O.); achmad.lefi@fk.unair.ac.id (A.L.); budi.baktijasa@fk.unair.ac.id (B.B.D.); bsp49@fk.unair.ac.id (B.S.P.); 2University Medical Center Groningen, 9713 GZ Groningen, The Netherlands; f.f.alkaff@umcg.nl

**Keywords:** COVID-19, hs-troponin, global longitudinal strain, mechanical ventilation, mortality

## Abstract

**Background.** Myocardial involvement among critically ill patients with coronavirus disease 2019 (COVID-19) often has worse outcomes. An imbalance in the oxygen supply causes the excessive release of pro-inflammatory cytokines, which results in increased ventilation requirements and the risk of death in COVID-19 patients. **Purpose.** We evaluated the association between the hs-troponin I levels and global longitudinal strain (GLS) as evidence of myocardial involvement among critical COVID-19 patients. **Methods.** We conducted a prospective cohort study from 1 February to 31 July 2021 at RSUD Dr. Soetomo, Surabaya, as a COVID-19 referral center. Of the 65 critical COVID-19 patients included, 41 (63.1%) were men, with a median age (interquartile range) of 51.0 years (20.0–75.0). Subjects were recruited based on WHO criteria for severe COVID-19, and myocardial involvement in the form of myocarditis was assessed using CDC criteria. Subjects were examined using echocardiography to measure the GLS, and blood samples were taken to measure the hs-troponin. Subjects were then followed for their need for mechanical ventilation and in-hospital mortality. **Results.** Severe COVID-19 patients with cardiac injury were associated with an increased need for intubation (78.5%) and an increased incidence of myocarditis (50.8%). There was a relationship between the use of intubation and the risk of death in patients (66.7% vs. 33.3%, *p*-value < 0.001). Decreased GLS and increased hs-troponin were associated with increased myocarditis (*p* values < 0.001 and 0.004, respectively). Decreased GLS was associated with a higher need for mechanical ventilation (12.17 + 4.79 vs. 15.65 + 4.90, *p*-value = 0.02) and higher mortality (11.36 + 4.64 vs. 14.74 + 4.82; *p*-value = 0.005). Elevated hs-troponin was associated with a higher need for mechanical ventilation (25.33% vs. 3.56%, *p*-value = 0.002) and higher mortality (34.57% vs. 5.76%, *p*-value = 0.002). **Conclusions.** Critically ill COVID-19 patients with myocardial involvement and elevated cardiac troponin levels are associated with a higher need for mechanical ventilation and higher mortality.

## 1. Introduction

COVID-19 is a worldwide pandemic with a mortality rate approaching ten percent in Indonesia [1]. Fever, cough, and dyspnea are common symptoms of COVID-19, which are exacerbated in the elderly and other comorbidities, such as chronic obstructive pulmonary disease or heart disease [2]. Cardiac injury, as one of the cardiac manifestations that exacerbate the systolic and diastolic functions, has potentially worsened the condition of COVID-19 patients and caused mortality [3].

Acute myocarditis in COVID-19 patients, which is associated with immunologic responses and cytokine storms, destroys the myocardium, which is characterized by an increase in a specific marker for cardiac enzymes: high-sensitive troponin I (hs c-TnI) [3,4,5].

Hue et al. reported that acute myocarditis also causes heart failure accompanied by an increase in NT-proBNP associated with a fatal outcome, a high risk of death for COVID-19 patients [6]. The adaptive response appears due to the cardiac burden in patients with COVID-19 infection. Echocardiography is an imaging modality used in patients with suspected cardiac injury [7,8].

Examination of several cardiac-marker enzymes are used to establish a diagnosis and predict the prognosis of patients with COVID-19 [9,10]. This study aims to determine the ability of hs c-TnI and the NT-proBNP level as non-invasive prognostic tools, and to predict the outcome of heart injury in COVID-19 patients without a prior history of heart disease.

## 2. Methods

### 2.1. Data Source

Primary data in this study were obtained through laboratory examinations measuring cardiac markers and echocardiography to assess the GLS in adult COVID-19 patients, which was performed by a certified cardiology specialist at Dr. Soetomo General Hospital, Surabaya.

### 2.2. Study Population and Design

An observational analytic study with a single-center prospective cohort design was used. Research and treatment of this patient was carried out at Dr. Soetomo General Hospital, Surabaya, from 1 July to 31 December 2020. The population in this study were all COVID-19-positive patients aged 18–90 years with clinical manifestations. WHO diagnostic criteria were used in the laboratory results.

### 2.3. Data Collection

Data were collected by total sampling from all included patients in a prospective cohort. All patient data were examined clinically apart from the research team independently. Primary data were obtained from interviews, physical examinations, laboratory tests, and echocardiography.

### 2.4. Statistical Analysis

Data were analyzed using inferential statistical analysis and were tested for normality using the one-sample Kolmogorov–Smirnov and Shapiro–Wilk tests. The independent *t*-test followed by linear regression statistical tests were used if the data were normally distributed, and the Wilcoxon Mann–Whitney test followed by polynomial regression statistical tests were used if the data were not normally distributed. The gender, blood type, ethnicity, and type of financing were analyzed using the χ2 and Fisher exact tests, followed by logistic regression. Multivariate analysis was carried out if there were significant variables. All analysis results were presented as narratives, tables, and graphs. Statistical analysis was performed using SPSS software version 25.0.0 for Windows, 2017 (Armonk, NY, USA: IBM Corp.).

## 3. Results

This research was conducted in the Intensive Care Unit (ICU) of Dr. Soetomo General Hospital, Surabaya, from 1 February to 31 July 2021.

### 3.1. Baseline Characteristics of Subjects

A total of 65 severe COVID-19 patients with HFpEF who met the inclusion and exclusion criteria were included in this study. Based on the clinical, echocardiographic, and laboratory criteria, 33 subjects were diagnosed with myocarditis. The length-of-stay, GLS, and LAVI values were normally distributed. Subjects had a median age of 56 years, and 63.1% were male. Most subjects had histories of hypertension (69.2%) and diabetes (60%). Significant increases in mortality were associated with increased hs-troponin levels (*p* = 0.011), increased procalcitonin levels (*p* = 0.001), increased serum creatinine (*p* < 0.001), a decreased mean GLS (*p* = 0.005), an increased degree of diastolic dysfunction (*p* = 0.003), the need for mechanical ventilation (*p* < 0.001), and the incidence of myocarditis (*p* = 0.002). Increased body mass index and NT-proBNP were also associated with increased mortality (*p* = 0.044 and *p* = 0.002, respectively). The baseline characteristics are presented in Table 1.

### 3.2. Subanalysis of Myocardial Injury and Severe COVID-19 Outcome

The bivariate analysis showed an association between increased NT-proBNP and hs-troponin-I levels and mortality (Figure 1).

Myocardial injury was characterized via cardiac-marker increases, particularly the NT-proBNP and hs-troponin. Bivariate analysis was conducted to evaluate the association of the two laboratory parameters with the outcomes of severe COVID-19 patients. There was a significant relationship between increased NT-proBNP and the degree of diastolic dysfunction (*p* < 0.001). Elevated hs-troponin was also associated with the need for mechanical ventilation and a higher degree of diastolic dysfunction (*p* = 0.044 and *p* = 0.001, respectively) (Table 2 and Figure 2 and Figure 3).

An analysis of the laboratory parameters using the receiver operating characteristic (ROC) showed that NT-pro-BNP, hs-troponin, and procalcitonin were reliable in predicting the mortality outcomes in patients with severe COVID-19 (NT pro-BNP: AUC = 0.720; *p* = 0.002; 95% CI = 0.595–0.844; hs-troponin: AUC = 0.722; *p* = 0.002; 95% CI = 0.599–0.846; procalcitonin: AUC = 0.731; *p* = 0.001; 95% CI = 0.607–0.856) (Figure 4).

A correlative analysis of the laboratory parameters using the Spearman test showed a moderate correlation between the NT-proBNP and hs-troponin (r = 0.412; *p*-value < 0.001) (Table 3). There was a weak correlation between increased hs-troponin and a shorter length of stay (r = 0.287, *p* = 0.02), but the correlation between the increased NT-proBNP and length of stay was not significant (r = 0.230, *p* = 0.065).

### 3.3. Subanalysis of Echocardiography Parameters and Severe COVID-19 Outcome

The echocardiographic assessment showed that the mean GLS value and degree of diastolic dysfunction were associated with mortality in COVID-19 patients. A subanalysis showed that a decrease in the mean GLS was associated with an increased need for mechanical ventilation (Table 3 and Figure 5). In addition, the receiver operating characteristic (ROC) analysis showed that the GLS might be able to predict mortality in severe COVID-19 patients (AUC = 0.700; *p*-value = 0.006; 95% CI = 0.568–0.833) (Figure 6).

### 3.4. Subanalysis of Clinical Characteristics and Myocarditis in Severe COVID-19

The diagnosis of myocarditis was based on clinical, echocardiographic, and laboratory criteria. Subjects suffering from myocarditis were associated with male gender, older age, a history of hypertension, and the incidence of acute kidney injury (AKI) during treatment. The occurrence of myocarditis was associated with increased creatinine serum (*p* < 0.001), increased NT-proBNP (*p* < 0.001), increased hs-troponin (*p* < 0.001), a decreased mean GLS (*p* = 0.004), and the need for mechanical ventilation (*p* < 0.001) (Table 4). Additionally, patients with diastolic dysfunction were associated with higher mortality (*p* = 0.032), increased incidence of myocarditis, including increased mitral velocity (MV grade, *p* = 0.047), and the occurrence of tricuspid regurgitation reflecting increased left atrial pressure (TR Vmax, *p* = 0.029, and TR maxPG, *p* = 0.026).

### 3.5. Subanalyses of Myocarditis Clinical Characteristics and Mortality

Subanalyses were performed to assess the clinical characteristics of the myocarditis subjects and mortality (Table 5). Myocarditis patients with diastolic dysfunction were associated with increased mortality (*p* = 0.005).

## 4. Discussion

### 4.1. Clinical Characteristics of the Research Subjects

COVID-19 patients are differentiated into five degrees of severity by the WHO based on complaints and clinical signs. There are two host responses to COVID-19 infection: the immune defense-based protective phase, and the inflammation-driven damaging phase. In the first phase, the focus is on improving the immune response. In contrast, in the second phase, efforts are made to suppress the excess immune response [11]. Increased inflammatory markers are obtained to moderate, severe, and critical degrees. The increase in such inflammatory markers occurs as a host immune response to the virus, which may be accompanied by acute myocardial injury [11,12,13].

Our research data show that patients with higher body mass indexes (BMIs) have better survival abilities. This condition is called the “obesity survival paradox (OSP).” This tendency toward bad outcomes is based on the fact that obesity is associated with various cardiovascular comorbidities. However, further studies are needed on the relationship between obesity and mortality in severe COVID-19 patients. The OSP in pneumonia was generally elaborated in a meta-analysis study conducted by Nie et al. in 2014 [14]. There are three explanations for the inverse correlation of obesity with mortality in pneumonia. First, obese patients tend to be aware of cardiovascular risks and perform adequate therapy. Second, tumor necrosis factor-alpha (TNF-a) is a pro-inflammatory cytokine that is important in inflammatory and immune responses. Fatty tissue produces TNF-a receptors, indicated by a decrease in the pneumonia severity in the obese group. Third, pneumonia patients with normal weights may not have good metabolic abilities to cope with increased catabolic stress [12,14].

The coronavirus binds to target cells through the ACE2 receptor, a homolog of angiotensin-converting enzymes, and converts angiotensin II into angiotensin I. It also reduces vasoconstriction mediated by the RAAS and plays a pro-inflammatory role in angiotensin II. The binding of SARS-CoV-2 to the ACE1 receptor results in a series of post-receptor signal changes that result in vasoconstriction, pro-inflammatory response, and endothelial dysfunction and affect myocardial injury and pro-thrombotic processes. The mechanisms mediated by the ACE2 receptor are related to the effects of hypertension, diabetes mellitus, and a history of CHD on more severe manifestations of COVID-19 [14,15].

Hypertension and DM are the most common comorbidities found in COVID-19 patients [16]. This is aligned with the clinical demographic of our research subjects. The presence of hypertension in the population of our study subjects was higher compared to multicenter studies conducted in China and the United States of America, which were 24% and 35% [16,17,18]. The percentage of DM patients was also higher in our study than in a study conducted in China (9.7%, 95% CI: 7.2–12.2%) [18]. The difference in the percentage of comorbidities in our study could be due to the inclusion of patients with severe COVID-19. Wang, et al. stated that COVID-19 patients admitted to the ICU had a higher comorbidity rate than those who were not admitted to the ICU (72.2% vs. 37.3%) [19]. Patients needing intubation support were associated with an increase in mortality (*p* = < 0.001) [19], similar to another study that demonstrated that severe pneumonia is independently associated with ICU care, the use of mechanical ventilation, and mortality [12,20].

### 4.2. Subanalysis of Myocardial Injury and the Outcomes of the Patients

Increased concentrations of cardiac biomarkers (NT-proBNP and hs-troponin) are correlated with the severity of COVID-19 infection [9,10]. Improvements in other inflammatory responses, such as procalcitonin and CRP, have also been associated with increased mortality [12,20]. Based on the description of the basic data on the laboratory parameters, higher levels of NT-proBNP and hs-troponin were obtained in subjects with mortality output (NT-proBNP: 1380.50 vs. 238.10, *p* = 0.002; hs-troponin: 34.57 vs. 5.76, *p*-value = 0.001). Nevertheless, only an increase in hs-troponin exceeding the normal values is associated with increased mortality. Several clinical factors strongly influence some of the underlying factors, including the adjustment of the normal value of NT-proBNP. In addition, there was also an association of increased procalcitonin with mortality (*p* = 0.001).

There is an association between hs-troponin and an increased need for intubation and diastolic dysfunction (*p* = 0.002 and 0.001, respectively). Hs-troponin levels in circulation indicate a systemic inflammatory impact caused by COVID-19 infection; thus, hs-troponin can be used to predict major adverse cardiovascular events (MACEs) [12,20,21]. The mechanism of myocardial injury can also be seen from an imbalance in the oxygen demand, increased ventricular strains, direct myocyte trauma, and the response to increased catecholamines [12,20,21]. 

In sepsis conditions, increased hs-troponin is associated with LV dysfunction, as seen from the transthoracic echocardiography examination. Ventricular dilatation and stress to the LV wall are other causes of increased hs-troponin in sepsis conditions [22]. Studies conducted by Mehta et al. in 2004 showed that patients with increased hs-troponin were associated with regional wall motion abnormalities (RWMAs) (56% vs. 6%; *p* = 0.002), decreases in EF (46% vs. 62%; *p* = 0.04), and increased mortality (56% vs. 24%; *p* = 0.04). This also indicates the linkage of the inflammatory process underlying the increase in hs-troponin, which is also associated with alterations in the picture of diastolic dysfunction in patients [23].

The hs-troponin level can predict 30-day death, namely, an AUC of 0.81 (95% CI: 0.73–0.88). Other laboratory parameters also support this capability in predicting mortality (NT-proBNP: AUC = 0.80; 95% CI = 0.74–0.86; procalcitonin: AUC = 0.77; 95% CI = 0.70–0.84). The results of the AUC analysis on our study samples also showed the fairly good abilities of hs-troponin, NT-proBNP, and procalcitonin in predicting mortality. In addition, it supports its usefulness in predicting the need for mechanical ventilation.

Through the AUC test in our study, we obtained almost similar mortality prediction capabilities between hs-troponin, NT-proBNP, and procalcitonin. Our research shows that the AUC of NT pro-BNP is 0.720 (*p* = 0.002; 95% CI = 0.595–0.844) and that of hs-troponin is 0.722 (*p* = 0.002; 95% CI = 0.599–0.846). These values are similar to the result in the meta-analysis study conducted by Wibowo et al. in 2021: namely, the ability of hs-troponin to predict mortality with an AUC of 0.73 (0.69–0.77) [23,24].

Through Spearman correlation analysis, a correlation of the hs-troponin increase with the NT-proBNP increase was obtained in our study subjects with a moderate correlation. This supports that serial examination in conjunction with NT-proBNP examination can help analyze the prognoses of the existing biomarkers.

NT-proBNP is secreted in response to increased myocardial wall stress. The MMR process regulates this condition through pro-inflammatory molecules, such as lipopolysaccharides, interleukin 1, CRP, and cardiac troponin I, independent of the ventricular function. Elevated circulating natriuretic levels are associated with myocardial injury, inflammatory processes, and interaction with ACE2. In addition, its increase can also result from acute heart failure conditions [25,26].

Increased mortality was also observed to be significantly associated with an increase in procalcitonin (*p*-value = 0.001). An increase in procalcitonin is observed particularly in severe COVID-19 patients [26,27]. This condition can be observed in COVID-19 infection and sepsis. This was demonstrated in a study conducted by Firani and Priscilla (2022), which evaluated the significance of procalcitonin in predicting mortality (OR = 17.78, *p* = 0.001). The relationship between the increase in procalcitonin and the increase in hs-troponin could be related to the condition of increasing troponin, which can also be observed in conditions of sepsis [28].

A study conducted by Lippi and Plebani in 2020 showed an increase in procalcitonin occurring in 50% of severe COVID-19 infections and 80% of critical patients. Meanwhile, bacterial co-infection was only obtained in 20% of severe COVID-19 patients and in 50% of the critical degrees [29].

Our research data also show the levels of CRP (15.60 vs. 14.60), NLR (11.78 vs.9.86), and D-dimer (5240 vs. 4620), which are higher in the event group. Increased inflammatory markers in severe COVID-19 patients may be associated with cytokine storms in the second week of infection. In this condition, leukocytes become over-activated, which results in the release of cytokines. The relationship between increased neutrophils and decreased lymphocytes in COVID-19 infection is unclear. However, it is often associated with the involvement of neutrophil extracellular traps (NETs). This causes neutrophil infiltration into the pulmonary capillaries and results in organ damage and acute respiratory distress syndrome (ARDS). In comparison, the decrease in lymphocytes is caused by the expression of ACE2 receptors by lymphocytes, which results in apoptosis [27].

### 4.3. Subanalysis of the Echocardiographic Profile of Severe COVID-19 Patient Outcomes

GLS examination shows cardiac abnormalities in the early stages, with normal values of from −17% to −18% [8,23,24]. Systemic inflammation induced by COVID-19 can cause LV and RV disorders and result in heart failure [8]. The cardiovascular comorbidities result in a series of cellular changes, including changes in endocardial fibers that result in decreased GLS [8,30]. This also explains that while a decrease in GLS can describe myo-pericardial injury due to an acute process in COVID-19, other chronic conditions can also affect the decrease in GLS [8,30]. 

Our study excluded patients with a previous history of coronary heart disease or heart failure. We excluded echocardiographic findings indicating chronic LV function impairment characterized by regional wall motion abnormalities (RWMAs) and LV dilatation. Based on research conducted by Huang et al. (2022) [31], the pattern of acute LV dysfunction found in 22% of COVID-19 patients is mostly characterized by the absence of LV dilatation and a hypokinetic global pattern. Most of them show a picture that resembles septic cardiomyopathy: global hypokinetic without an increase in the E/A ratio.

Our findings showed that most patients had preserved LVEFs, non-dilated LVs, and normal E/A ratios. However, LVEF assessment is less able to show subtle myocardial dysfunction [32]. Therefore, in the assessment of the LV function, we included the GLS assessment, which was indicated by a decrease in the LV function in most of the samples, with a mean of 12.92 + 4.99. The E/A ratio value was low in 20% of the samples, and only 3% showed an increase (E/A ratio > 1.5). The finding of non-dilated LVs with the normal estimated LV filling pressure indicates acute secondary injury [33].

The study by Lairez et al. in 2021 showed no correlation between the decrease in GLS and the increase in the hs-troponin level [34]. The correlation analysis of GLS with hs-troponin using the Spearman method that we performed also showed no relationship between the GLS decrease and hs-troponin increase (r = 0.245; *p =* 0.05). However, the decrease in the GLS was associated with an increase in mortality (11.36 ± 4.64 vs. 14.74 ± 4.82; *p*-value = 0.005). The GLS examination also showed a fairly good ability to predict mortality (AUC = 0.700; *p*-value = 0.006; 95% CI = 0.568–0.833).

In addition, studies conducted by Bevilacqua et al. in 2021 state that a decrease in the LV GLS (*cutoff* −16.1%) can predict a decrease in the PaO_2_/FiO_2_ (*p*/F) ratio of <100 mmHg during maintenance, which marks the need for mechanical ventilation use (HR = 4.0; 95% CI = 1.4–11.1; *p* = 0.008) [35].

Our study shows a correlation between GLS decrease and increased intubation needs (12.17 ± 4.79 vs. 15.65 ± 4.90, *p =* 0.02). The AUC analysis also showed the ability of decreased GLS to predict intubation needs quite well (AUC = 0.707, *p =* 0.018, 95% CI = 0.514–0.873).

The cytokine storms in COVID-19 can induce distributive shock and cause acute respiratory distress syndrome (ARDS) due to COVID-19 pneumonia. This results in a decreased RV systolic function. Shortening measurements of longitudinal fibers as obtained via Tricuspid Annular Plane Systolic Excursion (TAPSE) are not sensitive enough to assess the RV involvement. Several factors influence changes in the RV systolic function, including the use of mechanical ventilation. Therefore, assessment with the RV GLS, the measurement of the RV dilatation, and a comparison of the RV end-diastolic area to the LV end-diastolic area (RVEDA/LVEDA) can be used as a reference [31,36]. Our study did not perform thorough measurements of the RV systolic function, so we did not include them in the discussion. The assessment we carried out was based on a decrease in the TAPSE scores found in 17% of the sample.

### 4.4. Analysis of Relationship of Clinical Characteristics to Incidence of Myocarditis

COVID-19 infection is associated with the incidences of myocardial injury, one of which is caused by myocarditis [5]. Myocarditis is an inflammatory process involving the myocardium [7]. The US Centers for Disease Control and Prevention (CDC) categorize myocarditis into probable and confirmed cases based on clinical manifestations and diagnostic features.

Making a diagnosis of myocarditis is not easy considering the heterogeneity of the clinical presentation and nonspecific echocardiographic findings. Echocardiographic findings of patients with myocarditis are indicated by global ventricular dysfunction, RWMAs, or diastolic dysfunction. Endomyocardial biopsy (EMB) examination is still the gold standard in confirming myocarditis. However, the invasiveness and low availability of EMB make its use less feasible in clinical practice. A systematic review study conducted by Urban et al., 2022 [37], showed that the application of EMB in the diagnosis of COVID-19 myocarditis was only carried out in 33% of cases. This makes cardiac magnetic resonance (CMR) examination the reference standard in diagnosing acute myocarditis. Its application is demonstrated by CMR’s ability to identify myocardial edema and inflammation non-invasively [32,37]. The establishment of the diagnosis of myocarditis uses MRI and/or biopsy; however, due to limitations, the use of additional diagnoses using CMR and biopsy could not be performed in our study. The classification of myocarditis in our study is based on the presence of one or more new clinical presentations or worsening (chest pain/suppression/discomfort, tightness, palpitations, or the occurrence of syncope), with ≥ 1 new findings from increased hs-troponin, an abnormal ECG picture, and abnormal heart function indicated by regional wall motion abnormality (RWMA) disorders through echocardiography, or, if CMR findings that support myocarditis are obtained without being accompanied by other causes of complaints, according to the Advisory Committee on Immunization Practices (ACIP).

A study by Maino et. al. in 2021 performed a cardiovascular magnetic resonance (CMR) examination on symptomatic COVID-19 patients. The examination showed cardiac involvement, edema, and myocardial scars in 58% of the patients through late gadolinium enhancement (LGE) [21]. In 2020, Knight et al. also conducted a study using CMR examination during early recovery to assess the presence, type, and extent of myocardial injury in troponin-positive COVID-19 patients (during treatment) [38]. Their study showed that abnormalities in CMR are common even though the heart function is generally normal. CMR indicates ischemic heart disease (17%) and suspicion of the myocarditis picture through LGE (38%), and it sometimes indicates multiple pathologies (ischemic and non-ischemic, 14%). In addition, the lack of edema in this group indicates that scars, such as those due to myocarditis, may be permanent [38].

An increase in hs-troponin alone cannot describe the incidence of myocarditis; therefore, its increase can also result from microangiopathy and myocardial infarction [39]. Biopsy examinations show diffuse monocular infiltration or high viral loads by the SARS-CoV-2 virus [40,41]. However, limited infrastructure does not allow biopsies or CMR examinations to be carried out. In our study, myocardial injury assessment was performed based on CDC criteria. This statement is supported by various studies that show hs-troponin as a biomarker that is very useful in assessing myocardial injury, as well as studies that show hs-troponin as a gold standard for the early diagnosis of cardiac complications with good clinical relevance [40,41]. Hs-troponin is even able to assess subclinical myocardial inflammation, which helps determine early treatment and assess the prognostics of cardiac complications in patients with COVID-19 [40].

Myocardial injury, as shown by an increase in the hs-troponin level, was obtained in 36% of patients who were hospitalized due to COVID-19 infection. However, in the case of fulminant myocarditis, normal hs-troponin is often obtained. Hence, there are restrictions on the use of hs-troponin as a single examination in diagnosing myocarditis. Subclinical myocardial dysfunction was reported in 79% of COVID-19 patients who underwent a strain-imaging examination using speckle-tracking echocardiography (GLS). Therefore, the use of these two parameters simultaneously needs to be analyzed in diagnosing myocarditis [41]. 

The pattern shown by the GLS examination can be a differentiating tool in patients with suspected acute myocardial infarction and myocarditis. Both conditions are often encountered in patients with COVID-19. Ischemic necrosis shows an endocardial strain that can extend to the epicardium that shows a transmural character. Meanwhile, abnormalities in myocarditis show a global or regional ischemic pattern that is limited to the epicardial or mid-wall area [32]. 

We conducted a follow-up analysis on the myocarditis group and found that there was an association between an increase in the hs-troponin level, a decrease in the GLS, and an increase in the degree of diastolic dysfunction and the incidence of myocarditis (*p* < 0.001, 0.004, and 0.032, respectively). The increased risk of death in the myocarditis group is supported by the impact of the myocardial injury on the cardiovascular system. Myocardial injury and decreased GLS can recognize both subclinical and acute myocardial dysfunction, which are associated with mortality events. Meanwhile, the evaluation of the two simultaneously supports the diagnosis of myocarditis. The myocardial injury underlying myocarditis also increases the filling pressure of the LV and may lead to diastolic dysfunction. Diastolic dysfunction contributes to an increase in the degree of aggravation and mortality in COVID-19 patients. An analysis of the myocarditis group showed that increased diastolic dysfunction was associated with increased mortality (*p* < 0.005).

The degree of diastolic dysfunction is an evaluation to assess the therapeutic response and predictors of HF-related hospitalization and mortality outcomes in patients with HFpEF according to the American Society of Echocardiography (ASE). However, no study has analyzed the degree of diastolic dysfunction in COVID-19 patients. Research conducted by AlJaroudi et al. in 2012 in the general population explains that diastolic dysfunction is an independent predictor of mortality. Improvements in diastolic dysfunction ≥ 2 showed better mortality outcomes. The degree of diastolic dysfunction illustrates the impact of the correlation between cardiovascular comorbidities and the cardiac response to COVID-19 infection. Stretching of the myocardium will occur due to a persistent increase in the LV filling pressure and result in LA dilatation, which, at an advanced stage, increases the LAVI. This explains that higher degrees of diastolic dysfunction are associated with the anatomical and functional impacts of the myocardium in compensating for changes in the LV charging pressure that occur as a result of myocardial injury in COVID-19 patients [42]. 

### 4.5. Study Limitation

This study is a single-center study; thus, further research is needed with a larger number of subjects and a multicenter study approach to provide an overview of the clinical characteristics of cardiac injury in patients with COVID-19 infection. The diagnosis of myocarditis using CMR and biopsy could not be performed in our study. However, CMR and biopsy examination may help to determine the definitive cause of the myocardial injury that occurs in severe COVID-19 patients.

## 5. Conclusions

Our study reveals a positive correlation between high hs-troponin levels and the need for intubation and diastolic dysfunction in patients with severe COVID-19. A higher NT-proBNP level was found with an increased degree of diastolic dysfunction in patients with severe COVID-19. Patients with a severe degree of COVID-19 with cardiac injury are associated with the need for intubation and the incidence of myocarditis. There was a correlation between the incidence of myocarditis and the risk of mortality in patients. High hs-troponin and NT-proBNP levels were found to increase the mortality in patients with severe degrees of COVID-19. This study also showed that hs-troponin and NT-proBNP have good abilities to predict the mortality of patients with severe COVID-19. However, this study showed no correlation between the increase in the NT-proBNP level and the duration of hospitalization. In addition, there was a weak correlation between the increase in the hs-troponin level and a short duration of hospitalization.

## Figures and Tables

**Figure 1 jcm-13-00352-f001:**
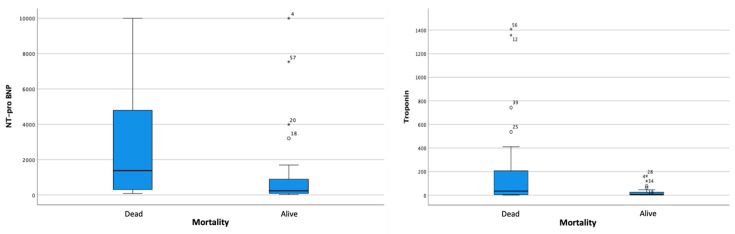
Boxplot analysis of relationship between increased (left side) NT-proBNP and (right side) hs-troponin and mortality.

**Figure 2 jcm-13-00352-f002:**
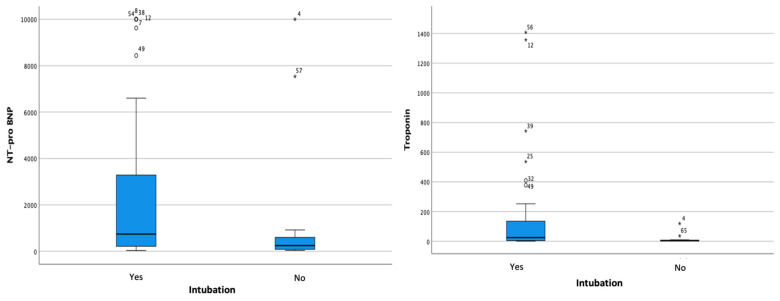
Boxplot analysis of increases in (left side) NT-proBNP and (right side) hs-troponin to intubation events.

**Figure 3 jcm-13-00352-f003:**
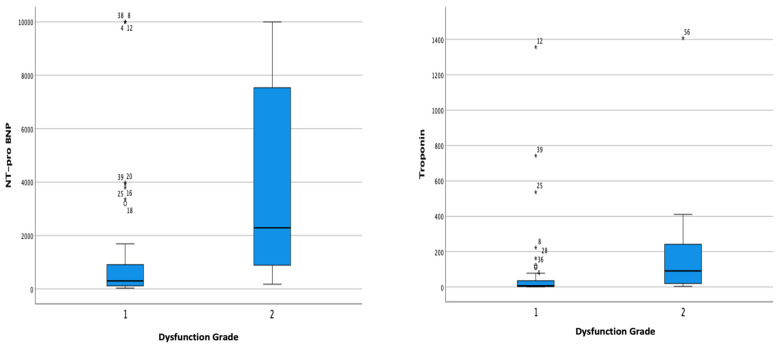
Boxplot analysis of relationship between increased (left side) NT-proBNP and (right side) hs-troponin and the degree of diastolic dysfunction.

**Figure 4 jcm-13-00352-f004:**
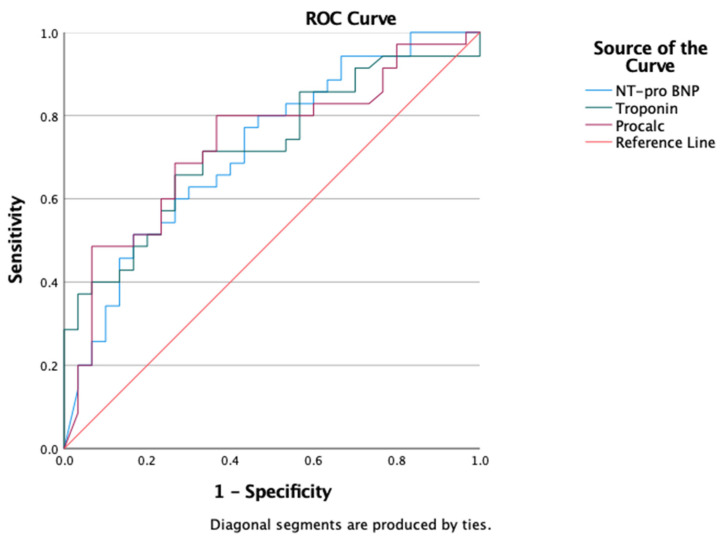
Analysis of receiver operating characteristic (ROC) of laboratory parameters on outcomes (mortality) of research subjects.

**Figure 5 jcm-13-00352-f005:**
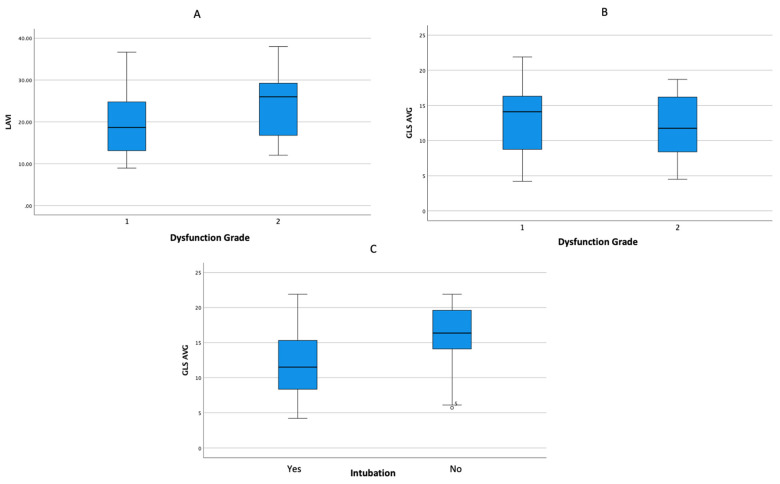
Boxplot analysis representing the following: (**A**) an increase in the LAVI is associated with an increase in the degree of diastolic dysfunction; (**B**) a decrease in the mean GLS is associated with an increase in the degree of diastolic dysfunction; and (**C**) a decrease in the mean GLS is associated with an increased need for intubation.

**Figure 6 jcm-13-00352-f006:**
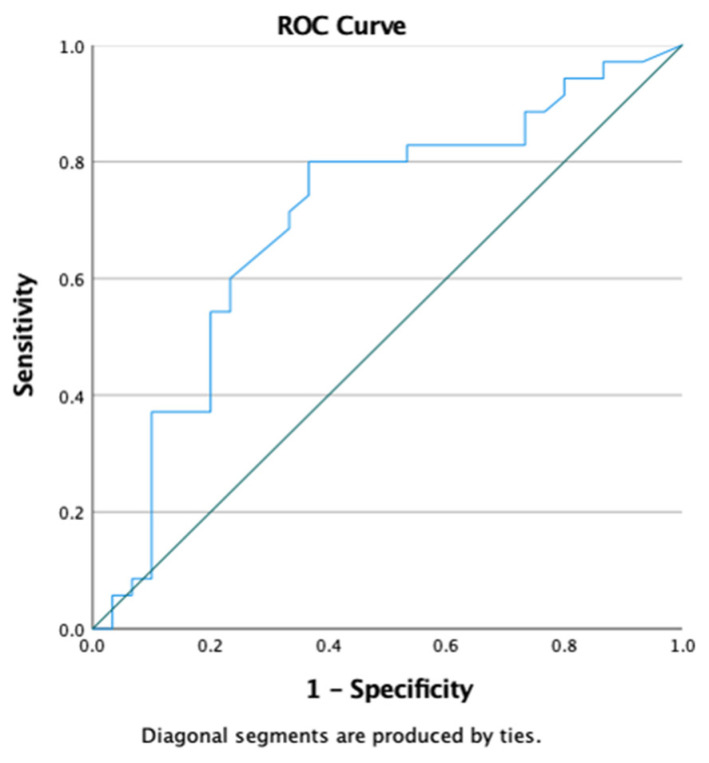
Receiver operating characteristic (ROC) analysis of GLS parameters on mortality. It is showed that GLS parameters could predict mortality (AUC 0.700, sensitivity 80%, specificity 64%).

**Table 1 jcm-13-00352-t001:** Baseline characteristics of subjects.

Variable	Total (n = 65)	Survived (n = 32)	Mortality (n = 33)	*p*-Value
Age	51 (20, 75)	46 (20, 75)	56 (25, 75)	0.085
Gender Male Female	41 (63.1%) 24 (36.9%)	20 (48.8%) 10 (41.7%)	21 (51.2%) 14 (58.3%)	0.579
Body Mass Index (kg/m^2^)	27 (17, 45)	28 (22, 37)	26 (17, 45)	0.044
**Past Medical History**				
Hypertension	45 (69.2%)	18 (40%)	27 (60%)	0.135
Diabetes Mellitus	39 (60%)	15 (38.5%)	24 (61.5%)	0.128
Pregnancy	8 (12.3%)	5 (62.5%)	3 (37.5%)	0.270
Acute Kidney Injury	23 (35.4%)	3 (13%)	20 (87%)	<0.001
**Laboratory Parameters**				
Serum Creatinine (mg/dL)	1.10 (0.10, 10.00)	0.80 (0.40, 3.70)	1.70 (0.10, 10.00)	<0.001
Increased NT-proBNP	54 (83.1%)	23 (42.6%)	31 (57.4%)	0.173
NT-proBNP (pg/mL)	581.90 (31, 10,000)	238.10 (32, 10,000)	1380.50 (82, 10,000)	0.002
Patients with Elevated Hs-Troponin-I	37 (56.9%)	12 (32.4%)	25 (67.6%)	0.011
Mean Level of Hs-Troponin-I (ng/L)	17.56 (1, 1407)	5.76 (2, 162)	34.57 (1, 1407)	0.002
Procalcitonin (ng/mL)	1.41 (0.01, 100)	0.65 (0.01, 100)	2.32 (0.05, 100)	0.001
CRP (mg/dL)	14.90 (0.36, 90.80)	14.6 (0.36, 28.90)	15.6 (0.50, 90.80)	0.571
NLR	10.85 (2.10, 44.80)	9.86 (3.87, 28.76)	11.78 (2.10, 44.80)	0.519
D-dimer	5030.00 (700, 25,230)	4620 (940, 22,300)	5240 (700, 25,230)	0.146
**Echocardiography Parameters**				
Biplane EF	66.00 (51, 85)	67.50 (53, 79)	65.00 (51, 85)	0.391
Average GLS	12.92 ± 4.99	14.74 ± 4.82	11.36 ± 4.64	0.005
LAVI	20.72 ± 7.31	20.05 ± 6.11	21.28 ± 8.24	0.503
Velocity MV E	0.59 (0.36, 0.92)	0.59 (0.41, 0.85)	0.58 (0.36, 0.92)	0.818
Average E/e’	6.99 (3.68, 14.00)	6.39 (4.62, 10.65)	6.47 (3.68, 14.00)	0.927
TR MaxPG	00.00 (0.00, 83.18)	0.00 (0.00, 56.66)	0.00 (00.00, 83.18)	0.018
TR Vmax	00.00 (00.00, 4.56)	0.00 (0.00, 3.76)	0.00 (0.00, 4.56)	0.019
Degree of Diastolic Dysfunction Grade 1 Grade 2	47 (72.3%) 18 (27.7%)	27 (57.4%) 3 (16.7%)	20 (42.6%) 15 (83.3%)	0.003
**Other Clinical Parameters**				
Length of Stay	19 ± 9	21 ± 9	18 ± 10	0.115
Intubation	51 (78.5%)	17 (33.3%)	34 (66.7%)	<0.001
ECMO	13 (20%)	7 (53.8%)	6 (46.2%)	0.534
Myocarditis	33 (50.8%)	9 (27.3%)	24 (72.7%)	0.002

DM, diabetes mellitus; AKI, acute kidney injury; CRP, C-reactive protein; NLR, neutrophil-to-lymphocyte ratio; EF, ejection fraction; LAVI, left atrial volume index; GLS, global longitudinal strain; TR, tricuspid regurgitation; ECMO, extracorporeal membrane oxygenator.

**Table 2 jcm-13-00352-t002:** Analysis of laboratory parameters in myocardial injury and the outcomes of patients with severe degrees of COVID-19.

Variable	Event	Non-Event	*p*-Value
Intubation	Intubation	Non-Intubation	
NT-proBNP Hs-Troponin Procalcitonin Serum Creatinine	742 (32–10,000) 25.33 (1–1407) 1.85 (0.05–100) 1.40 (0.10–10)	246.30 (37–10,000) 3.56 (2–120) 0.59 (0.05–100) 0.75 (0.50–1.10)	0.068 0.002 0.044 0.001
ECMO	ECMO	Non-ECMO	
NT-proBNP Hs-Troponin Procalcitonin Serum Creatinine	742 (32–10,000) 32.79 (2–1357) 2.01 (0.38–100) 0.80 (0.10–3.30)	419.20 (37–10,000) 16.44 (1–1407) 1.38 (0.01–100) 1.10 (0.40–10.00)	0.640 0.210 0.623 0.370
Diastolic Dysfunction	Grade 1	Grade 2	
NT-proBNP Hs-Troponin Procalcitonin Serum Creatinine	302.30 (32–10,000) 7.19 (1–1356) 1.20 (0.09–100) 1.00 (0.40–10.00)	2290.05 (178–10,000) 91.47 (4–1407) 1.97 (0.01–100) 1.30 (0.10–6.80)	<0.001 0.001 0.655 0.454

**Table 3 jcm-13-00352-t003:** Analysis of echocardiographic parameters on outcomes of patients with severe COVID-19.

Variable	Event	Non-Event	*p*-Value
Intubation	Intubated	Not intubated	
Length of Stay	19.71 ± 9.93	18.07 ± 7.05	0.567
GLS Average	12.17 ± 4.79	15.65 ± 4.90	0.02
LAVI	20.36 ± 7.74	22.02 ± 5.51	0.455
ECMO	ECMO	Non- ECMO	
Length of Stay	23.15 ± 9.88	18.40 ± 9.08	0.102
GLS Average	11.94 ± 4.71	13.17 ± 5.07	0.432
LAVI	22.21 ± 6.54	20.34 ± 7.50	0.415
Diastolic Dysfunction	Grade 1	Grade 2	
Length of Stay	20.49 ± 9.67	16.39 ± 7.99	0.115
GLS Average	13.35 ± 5.14	11.79 ± 4.49	0.261
LAVI	19.26 ± 6.77	24.52 ± 7.47	0.008

**Table 4 jcm-13-00352-t004:** Correlation between clinical characteristics and event of myocarditis in severe COVID-19 patients.

Variable	Non-Myocarditis (n = 32)	Myocarditis (n = 33)	*p*-Value
Age	49 (20, 75)	55 (25, 73)	0.753
Gender Male Female	21 (51.2%) 11 (45.8%)	20 (48.8%) 13 (54.2%)	0.675
Body Mass Index	27 (22.37)	27 (17, 45)	0.693
Past Medical History			
Hypertension	20 (44.4%)	25 (55.6%)	0.247
Diabetes Mellitus	21 (53.8%)	18 (46.2%)	0.362
Pregnancy	4 (50%)	4 (50%)	0.628
Acute Kidney Injury	8 (34.8%)	15 (65.2%)	0.085
Laboratory Parameters			
Serum Creatinine	0.80 (0.40, 10.00)	1.60 (0.10, 8.30)	<0.001
NT-proBNP	238.10 (32, 7536)	2524.40 (72, 10,000)	<0.001
Increased NT-proBNP	24 (44.4%)	30 (55.6%)	0.087
Hs-Troponin	3.56 (1, 36)	78.32 (2, 1407)	<0.001
Increased Hs-Troponin	6 (16.2%)	31 (83.8%)	<0.001
Procalcitonin	1.29 (0.01, 97.65)	1.92 (0.05, 100)	0.158
CRP	11.75 (0.36, 30.10)	15.70 (0.50, 90.80)	0.462
NLR	9.47 (3.21, 25.21)	13.22 (2.10, 44.80)	0.181
D-dimer	4440.00 (960, 25,230)	5720 (700, 22,300)	0.300
Echocardiography Parameters			
Biplane EF	67 (60, 79)	66 (51, 85)	0.669
MV Evel	0.58 (0.36, 0.81)	0.62 (0.42, 0.92)	0.047
Average E/E’	6.38 (3.77, 10.65)	6.76 (3.68, 14.00)	0.679
TR maxPG	0.00 (0.00, 56.66)	7.01 (0.00, 83.18)	0.026
TR Vmax	0.00 (0.00, 3.76)	1.32 (0.00, 4.56)	0.029
LAVI	19.99 ± 5.68	21.42 ± 8.63	0.436
Average GLS	14.70 ± 5.20	11.19 ± 4.16	0.004
Degree of Diastolic Dysfunction Grade 1 Grade 2	27 (57.4%) 5 (27.8%)	20 (42.6%) 13 (72.2%)	0.032
Other Clinical Parameters			
Intubation	19 (37.3%)	32 (62.7%)	<0.001
ECMO	6 (46.2%)	7 (53.8%)	0.804
Length of Stay	21 ± 9	18 ± 10	0.309
Mortality	11 (31.4%)	24 (68.6%)	0.002

DM, diabetes mellitus; AKI, acute kidney injury; CRP, C-reactive protein; NLR, Neutrophil to Lymphocyte Ratio; EF, ejection fraction; LAVI, left atrial volume index; GLS, global longitudinal strain; TR, tricuspid regurgitation; ECMO, extracorporeal membrane oxygenator.

**Table 5 jcm-13-00352-t005:** Relationship between clinical characteristics of myocarditis subjects and mortality in severe COVID-19 patients.

Myocarditis Variables	Survived (%)	Mortality (%)	*p*-Value
Total	9 (27.3%)	24 (72.7%)	-
Length of Stay	23 ± 11	16 ± 9	0.095
Intubation	8 (25%)	24 (75%)	0.273
ECMO	4 (57.1%)	3 (42.9%)	0.068
Demographic Parameters			
Age	44.0 (29, 63)	56.6 (25, 73)	0.392
Gender Male Female	7 (35%) 2 (15.4%)	13 (65%) 11 (84.6%)	0.216
Anthropometry			
Body Mass Index	28 (22, 36)	26.5 (17, 45)	0.766
Past Medical History			
Hypertension	5 (20%)	20 (80%)	0.117
Diabetes Mellitus	4 (22.2%)	14 (77.8%)	0.475
Pregnancy	1 (25%)	3 (75%)	0.705
Acute Kidney Injury	2 (13.3%)	13 (86.7%)	0.101
Laboratory Parameters			
Increased NT-proBNP	8 (26.7%)	22 (73.3%)	0.629
Increased HS-Troponin	8 (25.8%)	23 (74.2%)	0.477
Serum Creatinine	1.00 (0.70, 3.70)	0.10 (0.10, 8.30)	0.036
NT-proBNP	895.70 (72, 10,000)	3180.10 (120, 10,000)	0.179
Hs-Troponin	46.55 (2, 162)	135.99 (5, 1407)	0.079
Procalcitonin	0.70 (0.33, 100)	2.80 (0.05, 100)	0.437
CRP	14.30 (4.60, 23.30)	17.85 (0.50, 90.80)	0.538
NLR	13.22 (4.64, 28.76)	13.24 (2.10, 44.80)	0.890
D-dimer	5720 (940, 22,300)	5580 (700, 12,870)	0.984
Echocardiography Parameters			
Biplane EF	57 (53, 77)	65 (51, 85)	0.592
Average GLS	10.90 (4, 16)	10.40 (5, 22)	0.981
LAVI	19.36 ± 7.00	22.19 ± 9.18	0.409
MV E Vel	0.62 (0.42, 0.85)	0.63 (0.43, 0.92)	0.238
Average E/E’	5.87 (4.74, 10.53)	7.09 (3.68, 14.00)	0.619
TR Vmax	0.00 (0.00, 2.84)	3.10 (0.00, 4.56)	0.018
TR MaxPG	0.00 (00.00, 32.00)	38.37 (00.00, 83.18)	0.018
Diastolic Dysfunction Grade 1 Grade 2	9 (45%) 0 (0%)	11 (55%) 13 (100%)	0.005

DM, diabetes mellitus; AKI, acute kidney injury; CRP, C-reactive protein; NLR, Neutrophil to Lymphocyte Ratio; EF, ejection fraction; LAVI, left atrial volume index; GLS, global longitudinal strain; TR, tricuspid regurgitation; ECMO, extracorporeal membrane oxygenator.

## Data Availability

Data are available upon reasonable request.

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
