# Peer review of "Quantification of hs-Troponin Levels and Global Longitudinal Strain among Critical COVID-19 Patients with Myocardial Involvement"

_jcm, 2024, doi:10.3390/jcm13020352_

Round 1

Reviewer 1 Report

Comments and Suggestions for Authors

This study examined the relationship between hs-Troponin I levels and global longitudinal strain (GLS) as indicators of myocardial involvement in critical COVID-19 patients. The findings revealed that patients with myocardial involvement and elevated hs-Troponin levels had a higher need for mechanical ventilation and a greater mortality rate compared to a control group.

There are several critical issues that require attention:

1.     The manuscript needs a thorough English revision. For instance, "financing variables" in line 82 is unclear.

2.     The methods section should be enhanced, providing more information on cohort selection and data collection.

3.     Clarification is needed for why HFpEF was chosen in line 93. Authors should explain inclusion and exclusion criteria in the methods section and include a clear study flow chart.

4.     It's important to specify whether myocarditis diagnosis was performed using CMR (Cardiac Magnetic Resonance).

5.     The figures could be improved.

6.     The results section includes various analyses (e.g., ROC, Spearman) that are not adequately explained in the statistical section. Additionally, multivariable analyses are missing from the results section.

7.     It's worth considering that global longitudinal strain (GLS) may also be associated with other cardiac conditions, such as coronary artery disease (CAD) (please add relevant reference). In cases of myocardial damage without obstructive coronary artery disease, cardiac magnetic resonance is necessary (e.g., as demonstrated in 10.1016/j.jcmg.2023.05.016). Additionally, it's important to note that COVID-19 has been linked to reduced left ventricular (LV) and right ventricular (RV) GLS values (please add relevant reference).

Overall, the manuscript suffers from structural issues. A clear outline of inclusion/exclusion criteria, myocarditis diagnosis, data collection, and statistical analyses is essential for clarity and coherence.

Comments on the Quality of English Language

A thorough revision of English and a double check of abbreviations is required.

Author Response

Thank you for your constructive feedback. 

  1. We have revised several language and grammatical errors inside the manuscript according to Professional English Language Services
  2. We have added more information in the methodology
  3. We conducted a research among hospitalized patients with preserved ejection fraction before the beginning of the COVID-19 related myocarditis
  4. Unfortunately, we didn't have dedicated CMR in our hospital, but your suggestion are very good to shape the manuscript further
  5. We try to improve the figure and image quality
  6. We described the ROC curve as the analysis of laboratory parameters using the Spearman test showed a moderate correlation between NT-proBNP and hs-troponin (r=0.412; p-value <0.001) (table 3.3). There was a weak correlation between increased hs-troponin and shorter length of stay (r 0.287, p= 0.02), but the correlation between increased NT-proBNP and length of stay was not significant (r 0.230, p=0.065).

  7. Thank you for your suggestion, we have been considering cardiac MRI in the future as it can differentiate myocarditis, CAD, cardiomyopathy, ARVC, etc

Reviewer 2 Report

Comments and Suggestions for Authors

You must use the abbreviations, for example, at line 47 you abbreviate hs-troponin I as hs c-TnI, but several times you use hs-Troponin or hs-troponin or hs-Troponin I (We don't know if I or T is it in two first cites, we supose that it is I, but you don't write correct). You abbreviate Body Mass Index as BMC on table 1 but you don't use the abbreviation subsequently, the same with Acute Kidney Disease (AKD) and Diabetes Mellitus (DM). Please, review all the article.

In the other hand I suggest you that you review how you must cite the bibliography, for example, you write DOI but it is not correct within the reference citation rules of the journal. You must, also, write journals in short form.

Author Response

Thank you for your suggestion, I apologize for the mistake I have made. Actually hs-Troponin in the manuscript is similar to Hs-Troponin or hs-Troponin I, since all the reagent we used in high sensitive troponin I from ADVIA Centaur® Immunoassay Systems.

Thank you for your correction. Body mass index should be abbreviated with BMI, whilst Diabetes Mellitus with DM.

Thank you for your suggestion, we try to make it according to your reference citation rules of the journal.

Reviewer 3 Report

Comments and Suggestions for Authors

The author has already mentioned the limitation that there was no MRI or biopsy done, so it is hard to assess if myocarditis is due to inflammation or ischemia.

Obesity survival paradox from lines 208 - 219 is quite outdated ( 2014) and non-relevant to the current manuscript

The author should have mentioned that Troponin elevation could be due to associated CKD, PE, or demand ischemia. Other causes of Troponin elevation were ruled out.

Author Response

  • Thank you, it is our limitation that our hospital didn't have any dedicated cardiac MRI for COVID-19 patients
  • I got the newer references from one of the MDPI journal in 2023. Source: Dramé M, Godaert L. The Obesity Paradox and Mortality in Older Adults: A Systematic Review. Nutrients. 2023; 15(7):1780. https://doi.org/10.3390/nu15071780
  • Thank you for your kind suggestion, as elevated cardiac troponin could be due to the hypoxia and demand ischemic, sepsis, pulmonary embolism, or AKI / CKD.

Reviewer 4 Report

Comments and Suggestions for Authors

The study by Alsagaff et al. is original and interesting. However, there are several issues that need to be addressed with a revision:

- Introduction is too brief. Please provide a comprehensive background to justify the need for the study. I would suggest to discuss the results of the ECHO-COVID study (doi: 10.1007/s00134-022-06685-2) and its following post-hoc analysis (doi: 10.1007/s00134-023-07147-z) to describe the prevalence of systolic and diastolic dysfunction (and right ventricular involvement), in COVID-19 critically ill patients.

- Please specify which country Soetomo General Hospital Surabaya is in.

- "The population in this study were all 18-90-year-old". Please correct this sentence in order to specify that included patients were consecutive patients aged between 18 and 90 years old. Also, please explain why you excluded patients older than 90.

- Please better specify the inclusion and exclusion criteria. In particular, authors should specify which clinical manifestations of COVID-19 were taken into account (respiratory? every manifestation?)

- Please specify what you mean by "positive COVID-19 laboratory results according to WHO diagnostic criteria" or provide adequate reference to these criteria.

- Which guideline was used to assess diastolic function? Please specify.

- Please specify which laboratory tests were performed during the study.

- Did you calculate the sample size? How? Please describe.

- Authors should report the primary and secondary investigated outcomes in the methods section.

- Please provide ethical committee approval with code.

- Please include the low number of patients included as a possible limitation of the study.

Author Response

  • Thank you for your suggestion, we have extended our introduction section with the results of ECHO-COVID study
  • Soetomo General Hospital is in Indonesia
  • Thank you for your suggestion. As there are no limitation in our study, we collected all adults patients hospitalized with severe / critical COVID-19
  • Thank you for your suggesiton
  • Patients who got positive RT-PCR swab from the Authorized Laboratories for Testing COVID-19 according to the WHO
  • We used ACCF/ASE/ACEP/AHA/ASNC/SCAI/SCCT/SCMR 2008 handbook for assessing diastolic dysfunction
  • We used Soetomo Diagnostic Centre as Authorized Laboratories for Testing COVID-19 according to the WHO
  • Since we used total sampling, we did not perform any minimum sampling calculation.
  • Primary outcome was in-hospital mortality, whilst secondary outcomes were length of hospital stay, intubation and ECMO usage.
  • This study was authorized by the ethical committee of Dr. Soetomo General Academic Hospital, Surabaya, Indonesia (Reference number: 0133/KEPK/IV/2021)
  • Yes, limited number of samples is one of our limitation.

Round 2

Reviewer 1 Report

Comments and Suggestions for Authors

Kudos to the authors for the thorough revision, undeniably enhancing the manuscript's quality. Nevertheless, lingering uncertainties persist regarding certain previously raised points. Consequently, it is imperative to address these concerns in the manuscript's discussion/limitations section and meticulously update the bibliography accordingly.

It is noteworthy to consider that global longitudinal strain (GLS) may also exhibit associations with other cardiac conditions, such as coronary artery disease (CAD) (please refer to 10.1093/ehjci/jead046). In instances of myocardial damage unaccompanied by obstructive coronary artery disease, the necessity of cardiac magnetic resonance is underscored (as exemplified in 10.1016/j.jcmg.2023.05.016). Additionally, it is crucial to highlight that COVID-19 has been linked to diminished left ventricular (LV) and right ventricular (RV) GLS values (please discuss this extensively)

Comments on the Quality of English Language

Moderate English revision is still necessary. 

Author Response

We added the argument regarding GLS that may also exhibit associations with other cardiac conditions, such as CAD. We also describe how COVID-19 may diminished left ventricular (LV) and right ventricular (RV) GLS values in our discussion.

“The pattern shown by the GLS examination can be a differentiating tool in patients with suspected acute myocardial infarction and myocarditis. Both conditions are often encountered in patients with COVID-19. Ischemic necrosis shows an endocardial strain that can extend to the epicardial which shows a transmural character. Meanwhile, abnormalities in myocarditis show a global or regional ischemic pattern that is limited to the epicardial or midwall area.32”

“GLS examination shows cardiac abnormalities in the early stages, with normal values of -17% to -18%.8,23,24Systemic inflammation induced by COVID-19 can cause LV and RV disorders and result in heart failure.8 The cardiovascular comorbidities result in a series of cellular changes, including changes in endocardial fibers that result in decreased GLS.8,30 This also explains that while a decrease in GLS can describe myo-pericardial injury due to an acute process in COVID-19, other chronic conditions can also affect the decrease in GLS.8,30

“The cytokine storm in COVID-19 can induce distributive shock and cause acute respiratory distress syndrome (ARDS) due to COVID-19 pneumonia. This results in decreased RV systolic function. Shortening measurements of longitudinal fibers as obtained via Tricuspid Annular Plane Systolic Excursion (TAPSE) are not sensitive enough to assess RV involvement. Several factors influence changes in RV systolic function, including the use of mechanical ventilation. Therefore assessment with RV GLS, measurement of RV dilatation, and comparison of RV end-diastolic area to LV end-diastolic area (RVEDA/LVEDA) can be used as a reference.31,36 Our study did not perform thorough measurements of RV systolic function, so we did not include them in the discussion. The assessment we carried out was based on a decrease in TAPSE scores found in 17% of the sample.”

Reviewer 4 Report

Comments and Suggestions for Authors

After the first revision, the authors successfully addressed most of the comments I had provided. However, despite their reply, I verified that the introduction is still as brief as it was in the first version, and it was not modified in order to discuss the results of the ECHO-COVID study (doi: 10.1007/s00134-022-06685-2) and the following post-hoc analysis (doi: 10.1007/s00134-023-07147-z). Please discuss and add these 2 references.

Author Response

Our study excluded patients with a previous history of coronary heart disease or heart failure. We, therefore, excluded echocardiographic findings indicating chronic LV function impairment characterized by RWMA and LV dilatation. Based on research conducted by Huang et al (2022), the pattern of acute LV dysfunction found in 22% of COVID-19 patients is mostly characterized by the absence of LV dilatation and a hypokinetic global picture. Most of them show a picture that resembles septic cardiomyopathy, namely global hypokinetics without an increase in the E/A ratio.

Our findings showed that the majority of patients had preserved LVEF, non-dilated LV, and normal E/A ratio. However, LVEF assessment is less able to show subtle myocardial dysfunction (Wieczorkiewicz, 2022). Therefore, in the assessment of LV function, we included the GLS assessment which was indicated by a decrease in LV function in most of the samples, namely with a mean of 12.92 + 4.99. The E/A ratio value was low in 20% of samples and only 3% showed an increase (E/A ratio > 1.5). The finding of non-dilated LV with normal estimated LV filling pressure indicates acute secondary injury (Valenzuela, 2022).

The cytokine storm in COVID-19 can induce distributive shock and cause acute respiratory distress syndrome (ARDS) due to COVID-19 pneumonia. This results in decreased RV systolic function. Shortening measurements of longitudinal fibers as obtained via Tricuspid Annular Plane Systolic Excursion (TAPSE) are not sensitive enough to assess RV involvement. Several factors influence changes in RV systolic function, including the use of mechanical ventilation, therefore assessment with RV GLS, measurement of RV dilatation, and comparison of RV end-diastolic area to LV end-diastolic area (RVEDA/LVEDA) can be used as a reference (Huang et al, 2022; Huang et al, 2023). Our study did not perform thorough measurements of RV systolic function, so we did not include them in the discussion. The assessment we carried out was based on a decrease in TAPSE scores found in 17% of the sample.

Making a diagnosis of myocarditis is not easy considering the heterogeneity of clinical presentation and nonspecific echocardiographic findings. Echocardiographic findings of patients with myocarditis are indicated by global ventricular dysfunction, RWMA, or diastolic dysfunction. Endomyocardial biopsy (EMB) examination is still the gold standard in confirming myocarditis. However, the invasiveness of EMB and low availability make its use less feasible in clinical practice. A systematic review study conducted by Urban (2022), showed that the application of EMB in the diagnosis of COVID myocarditis was only carried out in 33% of cases. This makes cardiac magnetic resonance (CMR) examination the reference standard in diagnosing acute myocarditis. Its application is demonstrated by CMR's ability to identify myocardial edema and inflammation non-invasively (Urban, 2022; Wieczorkiewicz, 2022). It was not possible to apply EMB or CMR examinations in our study, so we used careful assessment using inflammatory markers and GLS assessment.

The pattern shown by the GLS examination can be a differentiating tool in patients with suspected Acute Myocardial Infarction and Myocarditis. Both conditions are often encountered in patients with COVID-19. Ischemic necrosis shows an endocardial strain that can extend to the epicardial which shows a transmural character. Meanwhile, abnormalities in myocarditis show a global or regional ischemic pattern that is limited to the epicardial or mid wall area (Wieczorkiewicz, 2022).
